



# EPR Study of NO radicals encased in modified open C60 Fullerenes

Klaus-Peter Dinse[1], Tatsuhisa Kato[2], Shota Hasegawa[2], Yoshifumi Hashikawa[2], Yasujiro Murata[2], and Robert Bittl[1]

[1]Freie Universität Berlin, Fachbereich Physik, Arnimallee 14, 14195 Berlin, Germany
[2]Institute for Chemical Research, Kyoto University, Uji, Kyoto 611-0011, Japan

**Correspondence:** Robert Bittl (robert.bittl@fu-berlin.de)

**Abstract.** Using pulsed EPR techniques, the low temperature magnetic properties of the NO radical being confined in a C60 derived cage are determined. It is found that the smallest principal $g$ value $g_3$, being assigned to the axis of the radical, deviates strongly from the free electron value. This behavior results from partial compensation of the spin and orbital contributions to the $g_3$ value. The measured value $g_3 = 0.77(5)$ yields information about the deviation of the locking potential from axial symmetry.
This 17 meV asymmetry is found to be quite small compared to the situation found for the same radical in polycrystalline or amorphous matrices ranging from 300 to 500 meV. The analysis of the temperature dependence of spin relaxation times resulted in a critical temperature of about 3.5 K, assigned to temperature activated motion of the radical with coupled rotational and translational degrees of freedom in the complicated 3-dimensional potential.

## 10 1 Introduction

In a series of recent publications, the Kyoto group has shown that it is possible to encapsulate small and even reactive molecules in a modified $C_{60}$ cage with tailored entrance and exit holes (Hasegawa et al., 2018a; Futagoishi et al., 2017; Hashikawa et al., 2018). Using such designer type open cages instead of closed structures creates a new route for the preparation of interesting compounds. The family of endohedral fullerenes having closed carbon cages like N@$C_{60}$ (Murphy et al., 1996),
He@$C_{60}$ (Saunders et al., 1994), and H$_2$@$C_{60}$ (Komatsu et al., 2005), as well as $C_{82}$ (Stevenson et al., 1999) based metallo-endohedrals can thus be expanded significantly. It has been shown that these new compounds can be stable under ambient conditions, allowing easy handling. If encapsulated molecules are paramagnetic, as in case of $^3O_2$ or $^2NO$, EPR is the method of choice for elucidating their properties. This allows determining not only the stationary spin Hamilton parameters but furthermore allows detecting of dynamic properties arising from internal dynamics or motion of the compound as a whole. In case
of La@$C_{82}$ for instance it was possible to conclude from an analysis of 2D EXCSY spectra that the metal ion is rigidly locked to the inside surface of the carbon cage (Rübsam et al., 1996). In the present case of encapsulated NO radical it was concluded from the broad variance of its principal g matrix values (Hasegawa et al., 2018a) that even at low temperatures the radical is not fixed to a particular site. It was remarkable that the very small value quoted for the axial component (Hasegawa et al., 2018a) of





0.225 deviates significantly from the value determined for NO radicals trapped in a single crystal host (Ryzhkov and Toscano,

2005), or NO radicals adsorbed in zeolites (Poeppl et al., 2000). This very small value of $g_3 = 0.225$, deduced by an analysis of a CW measurement, necessitated confirmation by pulse ESR experiments, better suited for the study of very broad spectra. So far, no nitrogen hyperfine data were reported, which might be important for a full characterization of the compound. It was the aim of the present study to obtain by multi-frequency EPR and ENDOR techniques a complete spin Hamiltonian parameter set for the encapsulated radical. Furthermore the anticipated effects of a non-spherical cage potential on the radical are explored.

In addition, the effects of structural modification of the cage are studied.

## 2   Experimental Part

### 2.1   Sample Preparation

NO radicals trapped in two slightly different modified C60 cages were studied, in the following abbreviated by NO@C60-OH1 and NO@C60-OH3, see Fig. 1. The notation indicates the modified exit ridge. NO@C60-OH1  (Hasegawa et al., 2018a) and

NO@C60-OH3 were prepared by following the literature (Hasegawa et al., 2018a) and the modified procedures are described in references (Hasegawa et al., 2018a; Hashikawa et al., 2018)].

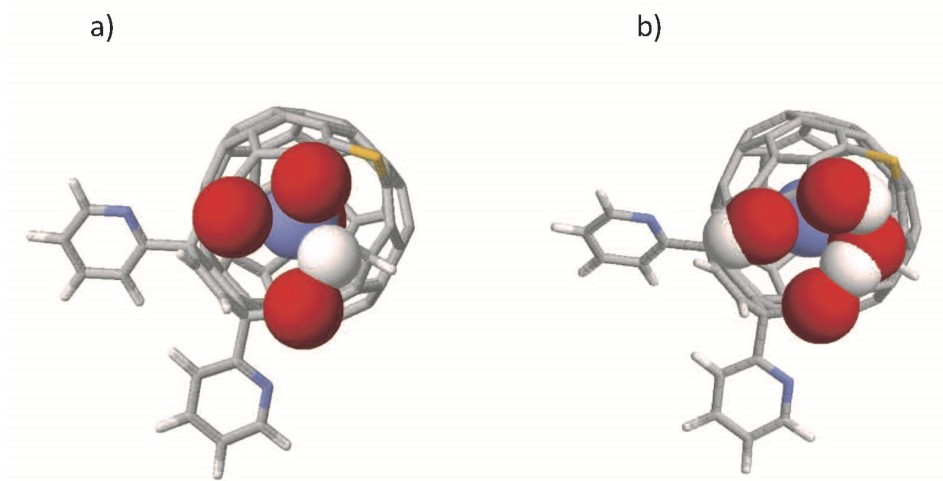

**Figure 1.** DFT optimized structures of a) NO@C60-OH1, and b) NO@C60-OH3. Oxygen (red) and hydrogen (white) atoms of the cage rim as well as nitrogen (blue) and oxygen (red) of the radical are indicated with van-der-Waals sized balls.

### 2.2   EPR Spectroscopy

For pulsed EPR and ENDOR measurements at different mw frequencies (3.4, 9.8, and 34 GHz), various setups were employed. Echo-detected 9.8 GHz EPR measurements at low temperatures were conducted with a Bruker ElexSys E680 setup equipped

with an Oxford CF930 helium cryostat using a Bruker MD4 Flexline ENDOR probe head. Field swept Echo-detected EPR





spectra (FSE) at 9.8 GHz were recorded using a two pulse "Hahn-echo" sequence (20-300-40 ns) at a temperature of 3.5 to 12 K, yielding absorption type spectra. FSE data at a microwave frequency of 3.4 GHz (S band) were obtained again using a Bruker ElexSys E680 system with additional S band accessory including a Bruker Flexline probe head with a split-ring resonator employing a pulse timing of 32-500-64 ns. Transient nutation measurements at 9.8 GHz were conducted applying a PEANUT pulse mw sequence with a $\pi/2$ pulse length of 8 ns, a delay time $\tau$ of 130 ns and a high turning angle (HTAx) pulse of 4096 ns. Phase inversion time within the high turning angle (HTAx) pulse was incremented by 2 ns starting with an initial inversion after 16 ns (Stoll et al., 1998). For 9.8 GHz HYSCORE measurements (Dinse et al., 2013), a $\pi/2$ pulse length of 16 ns and a delay time $\tau$ of 150 ns were used. ENDOR spectra were recorded applying either a Mims pulse sequence with $\pi/2$ pulses of 20 ns, delay time $\tau$ of 200 ns and a rf $\pi$ pulse length of 15 $\mu$s, or a Davies pulse sequence with pulse settings 40-30000-20-200 ns and a RF pulse length of 25 $\mu$s.

### 2.3 Quantum chemical calculations

Optimization of the structure of the compounds NO@C60-OH1 and NO@C60-OH3 has been performed using Gaussian (g16-A03) at the HPC center of FU Berlin. DFT calculations were performed using the 6-311++ basis set with UB3LYP exchange. Structures derived for the "up" orientation are depicted in Fig. 1. The difference in total energies for " up" and "down" orientations of the trapped radical was 22.6 meV for NO@C60-OH1 , somewhat larger than the value (8 meV) published earlier (Hasegawa et al., 2018a), which might be caused by use of a different basis set. For NO@C60-OH3 we calculated 40.2 meV.

## 3 Results and Discussion

### 3.1 Multi-Frequency EPR Data

EPR data published previously by Hasegawa *et al.* for NO@C60-OH1 were obtained in continuous wave (cw) mode at a microwave frequency of 9.56 GHz (Hasegawa et al., 2018a). The published $g$ matrix parameter set (see Table 1) obtained by spectral simulation of the cw spectrum is characterized by an extreme $g$ anisotropy. Values determined by FSE confirm the two larger $g$ matrix parameters, deviating however significantly with respect to the pseudo-axial $g_3$ parameter. Spectra measured at 3.45 and 9.76 GHz are depicted in Figs. 2 and 3, respectively.





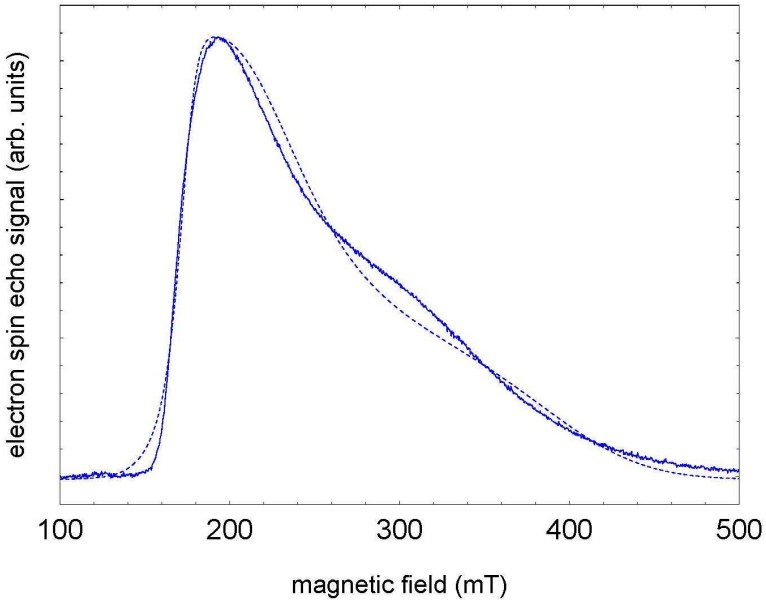

**Figure 2.** S-band (3.5 GHz) FSE spectrum of NO@C60-OH1 (10 mM / CS2, 5 K) with best fit. For fitting a set of nitrogen hyperfine tensor parameters was used, determined by ENDOR.

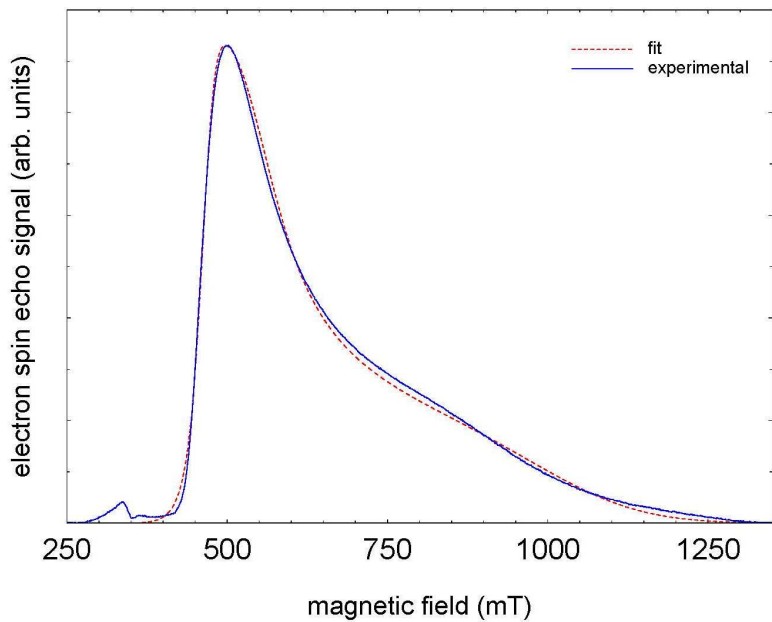

**Figure 3.** X-band (9.7 GHz) FSE spectrum of NO@C60-OH1 with best fit. For fitting a set of nitrogen hyperfine tensor parameters was used, determined by ENDOR.



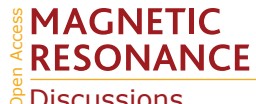

**Table 1.** Fit-determined $g$ matrix data of both compounds (gStrain fit data are listed in brackets). Previously published values (Hasegawa et al., 2018a) are shown for comparison. Level splittings deduced from the deviation of the pseudo-axial $g_3$ parameter from $g_e$ are also shown.

| sample | $\nu$ (GHz) | cw/FSE | $g_1$ | $g_2$ | $g_3$ | $\Delta$ (meV) |
|---|---|---|---|---|---|---|
| $NO@C_{60} - OH_1$ | 3.45 | FSE | 1.438(0.007) | 1.225(0.399) | 0.646(0.134) | 15.9 |
| $NO@C_{60} - OH_1$ | 9.76 | FSE | 1.482(0.002) | 1.350(0.275) | 0.679(0.182) | 16.9 |
| $NO@C_{60} - OH_3$ | 3.45 | FSE | 1.480(0.012) | 1.212(0.602) | 0.725(0.129) | 17.8 |
| $NO@C_{60} - OH_3$ | 9.76 | FSE | 1.527(0.002) | 1.422(0.287) | 0.767(0.173) | 19.7 |
| $NO@C_{60} - OH_1$ | 9.57 | cw | 1.488 | 1.320 | 0.225 | |

65      We quote no error margins, because a large $g$ strain value is obtained for the $g_3$ value using the fit routine (EasySpin (Stoll and Schweiger, 2006)). The pseudo-axial principal parameters $g_3$ = 0.631 and 0.679, respectively, are still found to be very small for the same compound, but rendering the $g$ matrix less anisotropic compared to the data in ref. (Hasegawa et al., 2018a). This discrepancy might be caused by the differing detection methods used. The FSE technique, displaying the EPR absorption directly, being best suited for recording broad unstructured spectra. For further confirmation of the $g$ matrix parameter set

70 determined by fitting the FSE spectra, we also performed a PEANUT experiment, probing the Rabi nutation frequency as function of $B_0$.

     Because of the rather large deviation of the $g_i$ parameters from the free electron value and the large anisotropy of $g$, a significant variation of the nutation frequency was expected as function of orientation. If by orientation selection a particular $g$ principal position is chosen, the two remaining g parameters in average determine the nutation frequency. As shown in Fig. 4,

75 all Rabi frequencies are smaller than the reference value determined by a standard coal sample and increase towards the high field spectral range. In the figure nutation frequencies are indicated, calculated using the values in Table 1. The agreement is quite convincing, and a very small $g_3$ parameter as deduced earlier can be excluded, since it would lead to a much smaller frequency in the perpendicular orientation of the radical. The small value of $g_3$ = 0.225 (Hasegawa et al., 2018a) is probably caused by overestimating the flat high field part of the cw spectrum in the simulation.



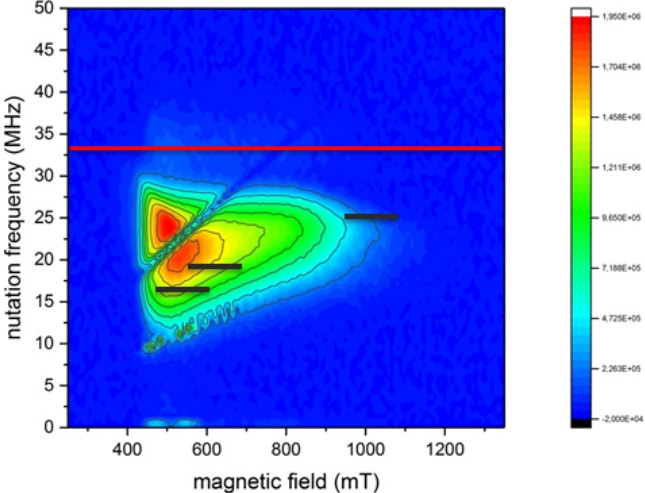

**Figure 4.** PEANUT spectrum of NO@C60-OH3 measured at 3.6 K. The red line indicates the reference frequency measured for a coal sample with isotropic $g = 2$.

Parameters determined for the NO@C60-OH3 compound are also listed in Table 1. Spectra are shown as Figs. A1 and A2 in appendix A. Also for this compound with slightly modified cage a similar set is observed, the fit parameters changing slightly towards larger values compared with those found for the OH1 compound. Even the slight difference in cage structure apparently is influencing the $g$ matrix values. However, no prominent features of anticipated magnetic interaction between encapsulated NO radicals within the intermolecular hydrogen-bonded dimeric triply hydroxylated C60 cages was observed.

It should be noted that $g$ matrix parameters of the encapsulated NO radical deviate much more from the free electron value $g_e = 2.0023$ value, comparing with data reported for situations when the radical is either trapped in a crystal ($g = $ (1.9740 (7), 1.9766 (7), 1.7175 (4))) (Ryzhkov and Toscano, 2005), adsorbed at the surface of metal oxides ($g = 1.97$, 1.97, 1.91) (Lunsford, 1968), or incorporated in a zeolite ($g = 2.001$, 1.996, 1.888)) (Poeppl et al., 2000). This clearly indicates that the orbital momentum of the radical is not fully quenched in the rather spherical capsule. Following the idea that partial quenching of the orbital angular momentum is accompanied by lifting of the degeneracy between the antibonding $^2\pi_x$ and $^2\pi_y$ orbitals, the energy splitting $\Delta$ between these orbitals can be estimated by the pseudo-axial value of the NO $g$ matrix (Ryzhkov and Toscano, 2005; Lunsford, 1968):

$$g_3 = g_e - 2\lambda L/(\lambda^2 + \Delta^2)^{1/2} \tag{1}$$

Here, $g_e$ is the free-electron $g$ value, $\lambda$ is the spin-orbit coupling constant for NO (123.16 cm$^{-1}$), $\Delta$ defines the crystal-field splitting of the $^2\pi_x$ and $^2\pi_y$ orbitals, and $L$ is a correction to the angular momentum along $z$ caused by the crystal field. $L$ is equal to 1 for a free molecule. A change in $L$ represents a modification of the molecular wave function by the crystal field. It should be noted, however, that in previous studies (Zeller and Känzig, 1967; Shuey and Zeller, 1967) no significant deviations from 1 were observed. The highly nonlinear dependence of $g_3$ on $\Delta$ is depicted in Fig. A3 (appendix A). Using Eq. (1), a level spitting of approximately 17 meV (200 K) is determined, The lifting of degeneracy leads to a deviation of the orbitals



from two fully circular symmetric angular momentum eigenstates with opposite momentum to two orthogonal elliptic orbitals not being angular momentum eigenstates, but with non-vanishing angular momentum expectation values. With a 200 K level splitting only one of the orbitals is occupied at 5 K and rotation of the molecule corresponds to transitions from one to the other eigenstate, which should be impossible due to the large level splitting. Nevertheless, the remaining angular momentum expectation value gives rise to very small $g_3$ value. The splitting is much less than values found for $^2$NO and $^2$O$_2^-$ trapped in
crystals, on surfaces or in zeolites, which are ranging from 300 to 500 meV.

   The $^2\pi_x$ and $^2\pi_y$ level splitting is of the same order of magnitude as the energy difference for the "up" and "down" orientation of the NO radical with respect to the cage opening calculated earlier (Hasegawa et al., 2018a) and also found in this study. For "up" / "down" axis reorientation a factor ten larger barrier was found. Considering the additional degree of freedom of hindered rotation about the axis of the radical with unknown transition barrier, this gives rise to a complicated 3-dimensional
orthorhombic potential energy surface. It is not surprising that under these conditions the EPR signal can be detected only at very low temperatures. Measuring the temperature dependence of the FSE signal (X-band) with respect to the decrease of a "standard" Boltzmann signal decrease of a stable $S = 1/2$ species in the sample, the apparent signal decay constant was determined as 3 K, shown in Fig. 5.

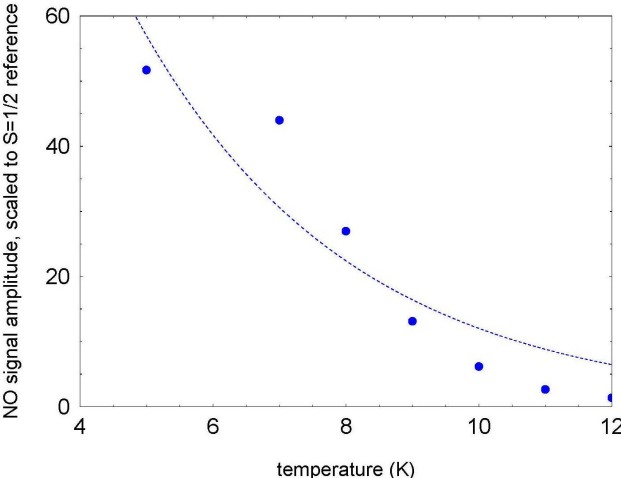

**Figure 5.** FSE-detected signal of the encapsulated NO radical (9.7 GHz, pulse distance 200 ns) as function of temperature. The signal intensity is scaled by the field separated signal of an unknown $S = 1/2$ radical, showing the regular Boltzmann dependence of signal intensity.

   The dramatic loss of signal intensity by a factor 50 in the narrow temperature range of 5 to 12 K is indicative for a decrease
of $T_2$. This was confirmed by measuring the 2-pulse echo decay constant $T_2^*$ at the peak signal position. Its temperature dependence could be fitted assuming exponential temperature dependence with a characteristic temperature of 3.9 K as shown in Fig. 6.



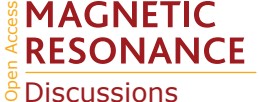

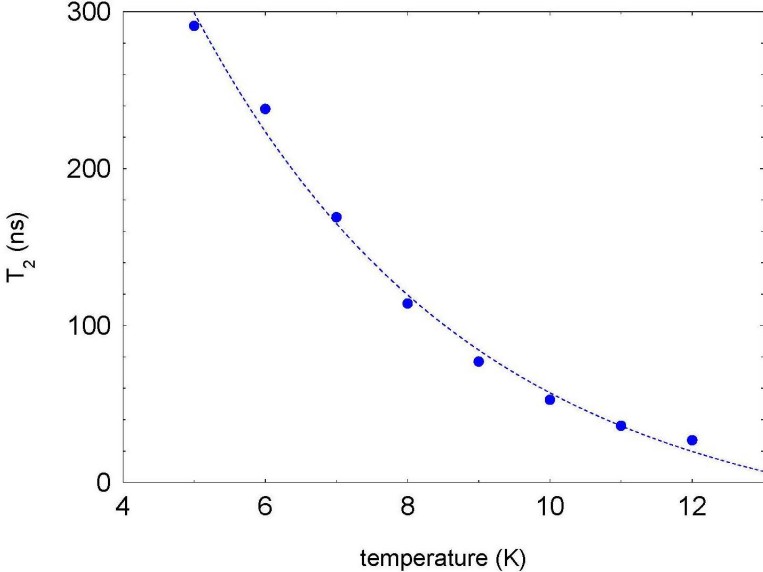

**Figure 6.** Temperature dependence of the spin echo decay constant measured at the FSE signal peak position (470 mT) of NO@C60-OH3 (2.5 mM in CS2). Fitting assuming exponential temperature dependence, the characteristic temperature is determined as 3.9 K.

Measuring the field dependence of $T_2^*$ at different temperatures, support the simple model of a restricted rotation. As shown in Fig. 7, at 5 K the $T_2^*$ values increase from 300 ns to 700 ns, when probing radicals changing from perpendicular to parallel orientation. This can be taken as evidence that small angle librations around the long axis are activated at this temperature, whereas long axis reorientations are still prevented at this temperature. In contrast, at 12.5 K this restriction is no longer valid, shortening the echo decay accordingly for the full field range.





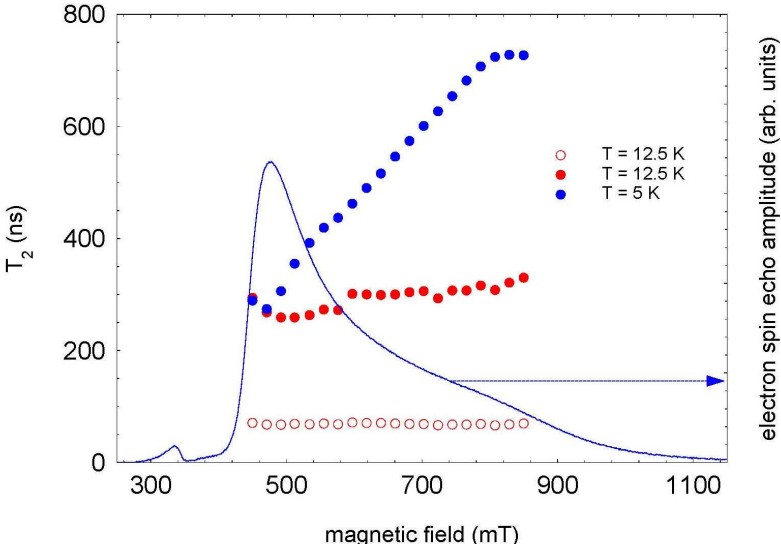

**Figure 7.** Field and temperature dependence of 2-pulse echo decay of NO@C60-OH3. The 5 K data set could be accurately fitted assuming single exponential decay; the 12.5 K data required a bi-exponential fit. Both components were of similar amplitude.

This hypothesis is also supported by the observation that $T_1$, determined by inversion recovery, also increases significantly when selecting radicals in parallel orientation (see Fig.8). This field dependence of $T_1$ leads even at 3.6 K to a noticeable change in the FSE pattern, if the pulse repetition time is not sufficiently long (see Fig. A4, appendix A).

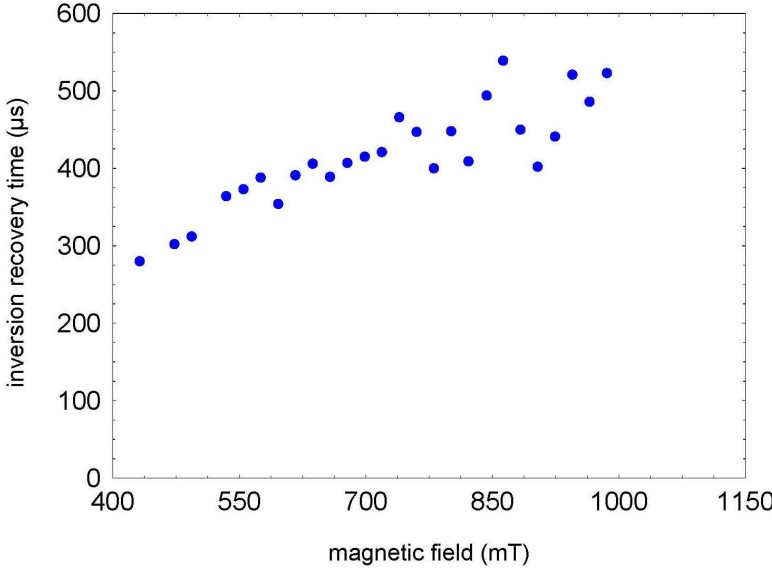

**Figure 8.** Field dependence of $T_1$ of NO@C60-OH3, measured using an inversion recovery pulse sequence at 3.6 K.





Loss of the cw EPR signal intensity at temperatures above 80 K was also reported in ref. (Hasegawa et al., 2018b). Since the cw signal intensity is not affected by $T_2^*$, the NO signal could be detected in cw mode up to 80 K (Hasegawa et al., 2018b) with a much smaller decrease from 5 K to 20 K than observed in our pulsed EPR study probing the echo signal with a 2-pulse sequence. The low critical temperature of 3.5 K (0.3 meV) (average value) has to compared to much larger values

found in the case of N@C60, and P@C60, in which a well-defined potential of spherical or axial symmetry leads to degenerate vibrational levels of the translational degree of freedom of encapsulated atoms in the range of 8 to 16 meV (Pietzak et al., 2002), respectively. The partially opened cage resembles the situation in the C70 cage by providing a nearly axial potential. Assuming that vibration along this preferred axis is lowest in energy and taking into account the larger mass of the radical, a vibrational eigenfrequency of about 5 meV for the center of mass (CM) of the radical would be expected, which is still

more than one order of magnitude larger than the experimental value. In contrast to encapsulated atoms, we have also to consider for the NO case a librational mode of the radical with respect to the cage axis. In a study of $H_2$ encapsulated in C60 or C70, the eigenstates of $H_2$ were determined numerically by invoking the appropriate 5-dimensional potential surface, describing translational and rotational degrees of freedom (Xu et al., 2009; Mamone et al., 2013). Lacking numerical values for the potential surface in our more complicated case, it is only possible to estimate typical values for the librational mode by

approximating the interconversion between up/down (its $z$ axis) of the radical axis in a potential well of 80 meV (645 cm$^{-1}$). According to Eq. (2), this gives rise to a characteristic energy of 1.8 meV (14.6 cm$^{-1}$), when approximating the potential by a harmonic function of amplitude A=40 meV/$\pi$rad. When including transverse degrees of freedom for axis reorientation, it is not likely that the characteristic mode energies might further be reduced thus matching the experimental value.

$$\omega = (A/2\Theta)^{1/2} \tag{2}$$

Here the moment of inertia of the radical is denoted by $\Theta$, and the energy difference for reorientation by $\pi$ radians is denoted by $A$.

## 3.2 ENDOR spectra

Orientation selective ENDOR spectra of NO@C60-OH1 were measured at 9.7 GHz. As depicted in Fig. 9, the center of lines shifts towards higher frequency, when changing the observation field position from lowest to highest edges of the absorption

pattern. A shift of the center of gravity of the ENDOR pattern is indicative for a dominant dipolar hfi, allowing simple determination of $A_i$ for the extreme field positions. For a determination of dipolar and quadrupolar hfi parameters observation field values at the low and high ends of the FSE spectrum were chosen, anticipating that g matrix and hfi tensor axes are collinear. Best ENDOR resolution is obtained at the low field edge, allowing determination of some hfi parameters by fitting, as shown in Fig. 10.



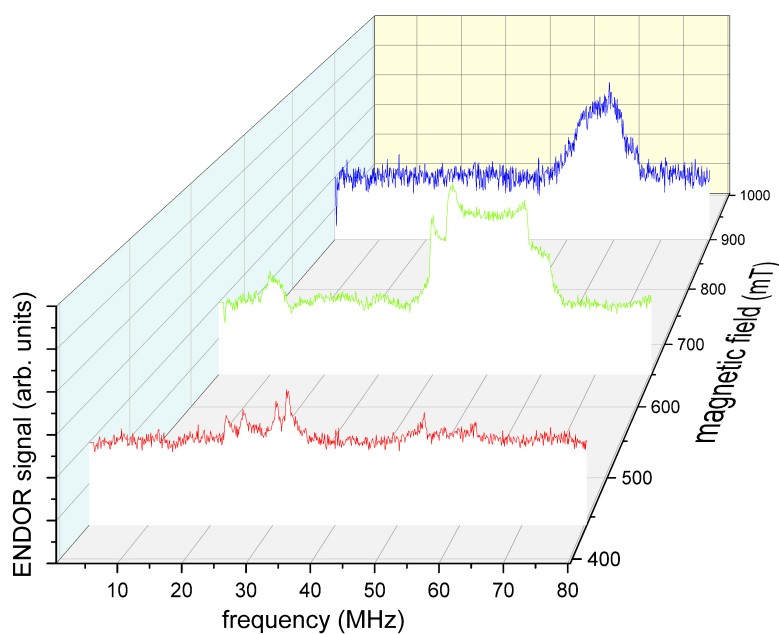

**Figure 9.** Davies ENDOR spectra of NO@C60-OH1 (10 mM in $CS_2$) measured at 5 K as function of $B_0$. Spectra are corrected with respect to different accumulation times for better comparison of spectral pattern. Pulse sequence used (40-30000-20-200-40 ns, 25 $\mu$s rf pulse) was identical for all spectra.

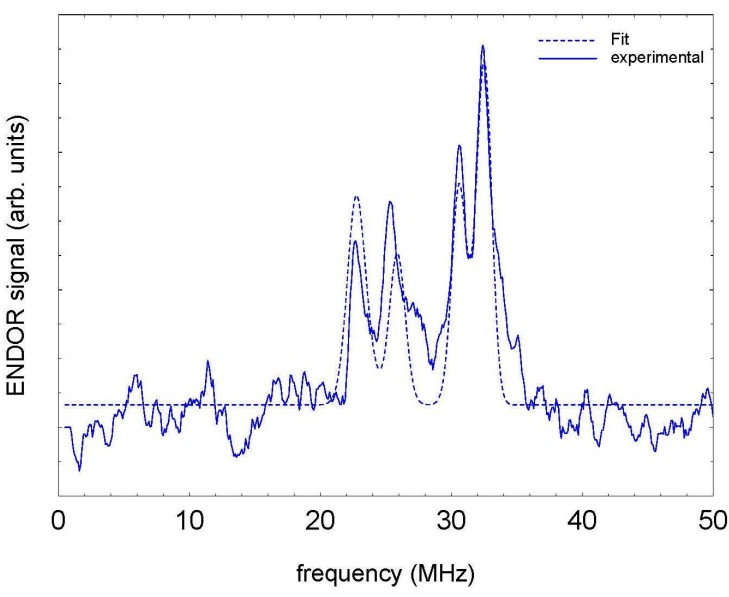

**Figure 10.** ENDOR spectrum of NO@C60-OH1 measured at 440 mT (T=5 K) using a Davies pulse sequence.





At this field position a consistent fit is obtained, by only fixing the nuclear Larmor frequency to its field-determined value. At the high field edge no line quartet is observed for this compound. The broad pattern, however, is consistent with the result of a spectral simulation, shown in Fig. 11, using a parameter set completed with the nqi parameter of NO@C60-OH3, being better resolved at the high edge of the ENDOR pattern. It should be noted that no simple pattern is expected for the intermediate field range because of significant g strain. For this reason fit values are only quoted assigned to $g_1$ and $g_3$ axes directions. No

information about the signs of hfi parameters can be deduced from the experimental spectra. The assignments given in Table 2 are tentatively made by invoking the calculated hfi constants (see Table 3). Although not being in very good quantitative agreement with the experiment, the calculated small isotropic hfi (+15 MHz) necessitates assignment of a negative sign to $A_1$. Lacking spectral resolution when probing at the high field edge due to the large $g_3$ strain, the center of gravity still gives a reliable value for the large dipolar hfi for both compounds. The absent spectral resolution, even when observing at the van

Hove singularities of the FSE spectrum, could result from a simultaneous presence of "up/down" configurations as observed in X-ray crystallography, with slightly different hfi parameters.

**Table 2.** Hyperfine parameters determined by fitting Davies ENDOR spectra measured under orientation selection conditions providing best resolution. For an assignment of signs see text.

| sample | $A_1$ (MHz) | $A_2$ (MHz) | $A_3$ (MHz) | $Q_1$ (MHz) | $Q_2$ (MHz) | $Q_3$ (MHz) |
|---|---|---|---|---|---|---|
| $NO@C_{60} - OH_1$ | -55.3 | N/A | +122.6 | 2.47 | N/A | N/A |
| $NO@C_{60} - OH_3$ | N/A | N/A | +124.1 | N/A | N/A | 1.1 |

**Table 3.** Hyperfine parameters calculated for NO@C60-OH1 and NO@C60-OH3 in their "up" configuration using Gaussian G16/A03 (G16/A03, B3LYP, 6-311++). The calculated values for the "down" orientation differ by less than 3%.

| sample | $A_1$ (MHz) | $A_2$ (MHz) | $A_3$ (MHz) | $Q_1$ (MHz) | $Q_2$ (MHz) | $Q_3$ (MHz) |
|---|---|---|---|---|---|---|
| $NO@C_{60} - OH_1$ | -25.5 | -23.5 | +90.2 | -1.48 | +0.22 | +1.26 |
| $NO@C_{60} - OH_3$ | -25.4 | -23.1 | +90.5 | -1.48 | +0.22 | +1.26 |





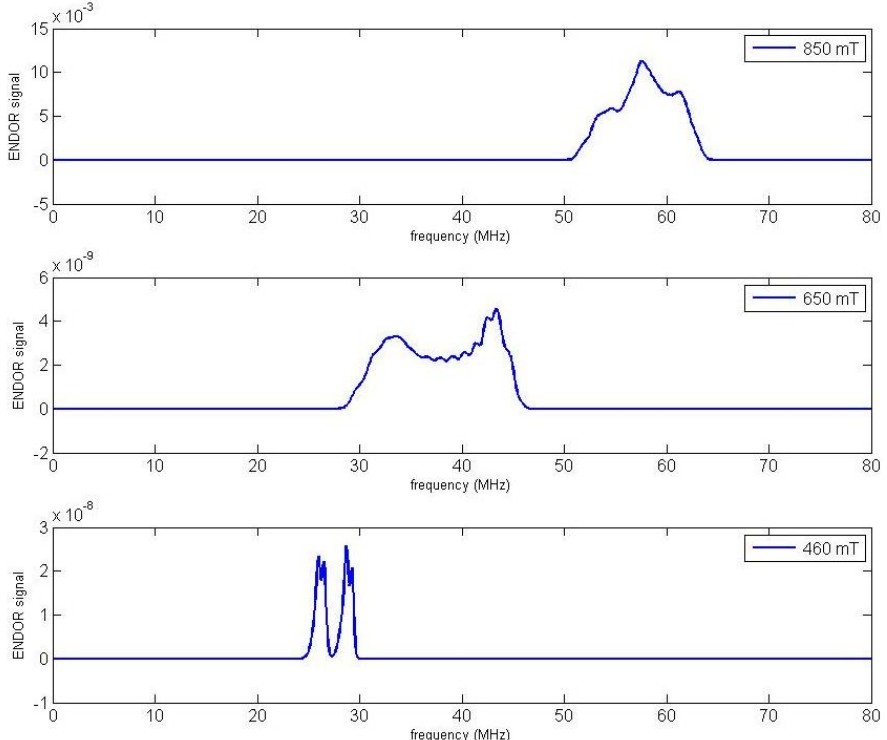

**Figure 11.** Simulated ENDOR spectra of NO@C60-OH1, using parameters listed in Table 2.

## 4 Conclusions

Using various EPR techniques, the spin Hamiltonian parameters for the encapsulated NO radical are determined. The radical, being confined in a $C_{60}$ derived cage, exemplifies the transition between a free molecule in isotropic potential and being fixed by a rigid confinement. The NO radical is particularly suited for such an investigation, since the $g$ factor of the free molecule in its $^2\Pi_{1/2}$ rotational ground state will change between zero (Mendt and Pöppl, 2015) to a $g$ matrix, in with all parameters are close to the free electron value for the rigidly localized radical (Chiesa et al., 2010). In case the axial molecular symmetry is maintained by the environment allowing free rotation about its axis, the $g$ parameter $g_3$, being assigned to the NO bond axis is predicted to vanish. The measured value $g_3 = 0.77(5)$ is indicative for an intermediate situation of the radical and yields information about the locking potential's deviation from axial symmetry. This 17 meV asymmetry as found here is quite small compared to the situation in polycrystalline or amorphous matrices ranging from 300 to 500 meV. The analysis of the spin relaxation times resulted in a critical temperature of about 3.5 K, assigned to temperature activated motion of the radical with coupled rotational and translational degrees of freedom in the complicated 3-dimensional potential provided by the cage.

Performing ENDOR the $^{14}$N hyperfine coupling parameters were determined. The experimental values are in fair agreement with predictions from a DFT calculation. The spectral resolution was not sufficient to discriminate between parameter sets expected for the Xray crystallography confirmed "up/down" configurations of the radical with respect to the hole of the cage.





Hfi as well as $g$ matrix parameters did not show any temperature dependence in the range of 3.5 to 12 K, in which a dramatic decrease of $T_2^*$ is observed. This indicates that the radical is localized, not allowing for excitation of rotational modes round its axis, which would modify the $g_3$ value. Apparently only low energy modes with small amplitude around its equilibrium

orientation are excited at these temperatures. It should be noted, however, that the accuracy of the data analysis is high enough to detect a small difference in $g$ parameters using cages with slightly modified openings. It will be interesting to see in the future, if advanced computational methods will be able to simulate $g$ matrix and hfi tensor data for this radical in such a complicated potential.

*Code and data availability.* Experimental data, processing information, and EasySpin simulations scripts will be made available upon re-

quest for reviewing and uploaded to the refubium.fu-berlin.de institutional repository prior to publication.

## Appendix A: Supplementary Figures

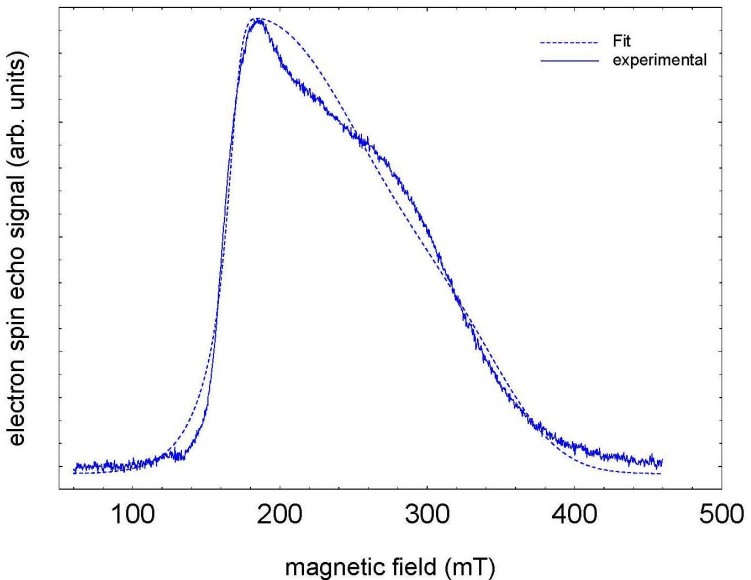

**Figure A1.** S-band (3.5 GHz) FSE spectrum of NO@C60-OH3 (T = 5K, 10mM/CS$_2$) with best fit.



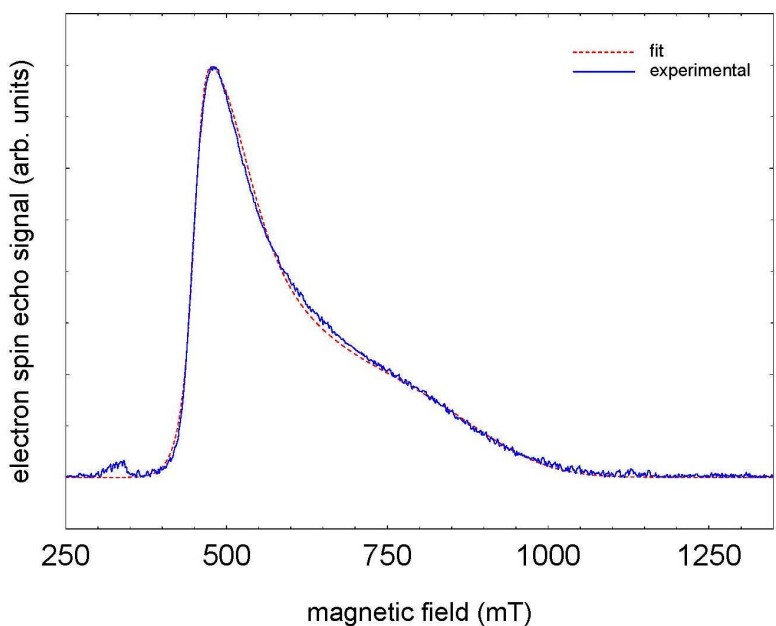

**Figure A2.** X-band (9.7 GHz) FSE spectrum of NO@C60-OH3 (T = 5 K, 2.5 mM/CS$_2$) with best fit.

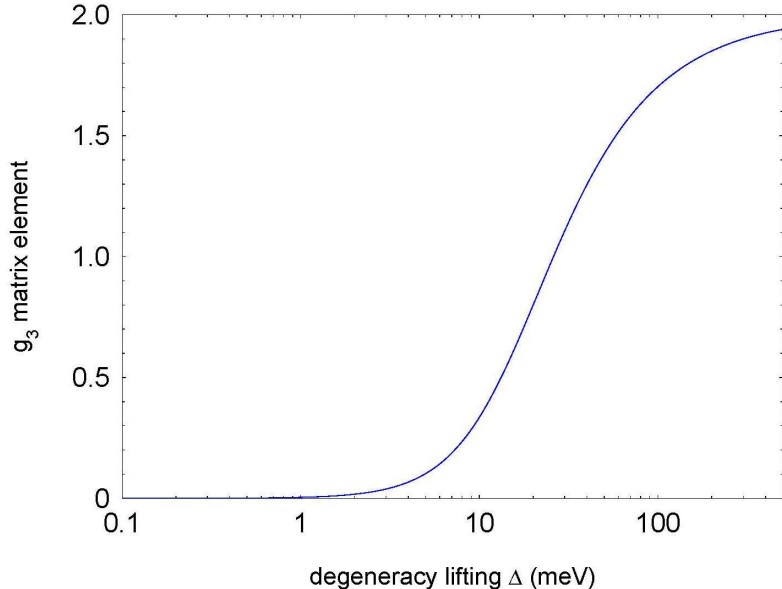

**Figure A3.** Dependence of the pseudo-axial $g_3$ matrix element of the NO radical as function of $^2\pi_x^*$ and $^2\pi_y^*$ level spitting.





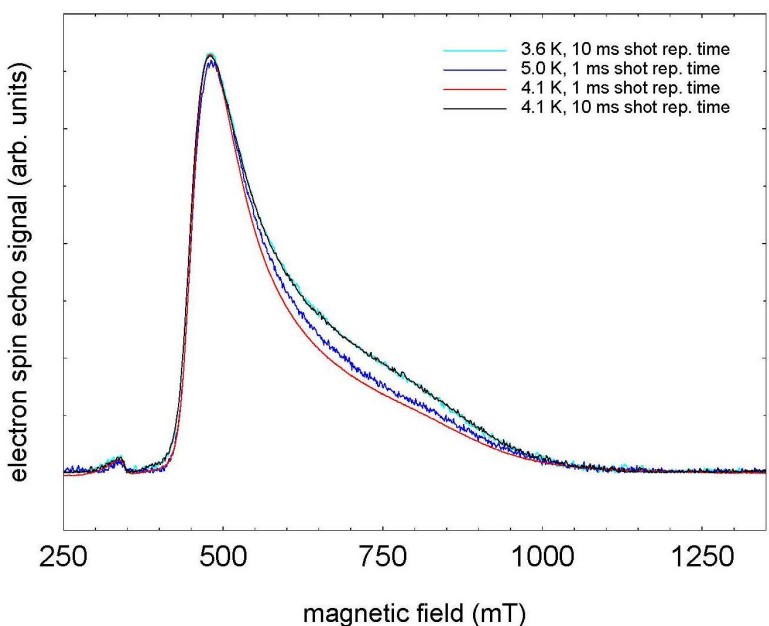

**Figure A4.** 9.7 GHz FSE spectra of NO@C60-OH3 measured at 3.6, 4.1, and 5 K. Using a rather short pulse repetition time (1 ms), the high field part of the spectrum is partially saturated. (For fit parameters see Table B1.)

## Appendix B: Supplementary Table

**Table B1.** Best fit parameters for the NO@C60-OH3 spectra measured at different temperatures (see Fig. A4).

|  | $g_1$ | $g_2$ | $g_3$ | $g_1 strain$ | $g_2 strain$ | $g_3 strain$ | linewidth (mT) |
|---|---|---|---|---|---|---|---|
| 5 K, 120 ns | 1.525 | 1.425 | 0.744 | 0.001 | 0.294 | 0.171 | 17.4 |
| 10 K, 120 ns | 1.521 | 1.420 | 0.717 | 0.001 | 0.348 | 0.131 | 12.6 |
| 10 K,300 ns | 1.505 | 1.419 | 0.699 | 0.001 | 0.432 | 0.139 | 13.2 |

*Author contributions.* Compounds were synthesized by SH, YH, and YM. EPR experiments were performed by KPD, TK, and RB. Data analysis was performed by KPD and RB, and the manuscript written by KPD with input from all authors.

*Competing interests.* The authors declare no competing interests

*Acknowledgements.* The authors thank Claudia Tait for helpful discussions and the HPC Service of ZEDAT (Freie Universität Berlin) for computing time.





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
