# Peer review of "EPR Study of NO radicals encased in modified open C60 Fullerenes"

_Magnetic Resonance, 2020_

## Referee Comment (RC1) · Malcolm Levitt (Referee) · 5 Jun 2020

This is a very interesting paper describing the EPR and ENDOR of a system in which single paramagnetic NO molecules are encapsulated in open-cage C60 fullerenes. A thorough analysis of the EPR data is given in terms of the g-tensor parameters and the ENDOR is analysed to obtain hyperfine coupling parameters to the 14N nucleus. The temperature-dependence of the EPR parameters and also relaxation times are analysed in terms of a postulated motional model of the encapsulated molecule. The paper describes fine experimental work on an exciting and novel physical system and is highly suitable for publication in MagRes. I have a few small proposals for possible improvements. 1. In the abstract, it is not very clear that the confining cage is not C60 but an open-cage variant (in fact two variants) of C60. This is important since the

symmetry of the confining potential has a very strong influence on the behaviour of the confined system. 2. In the introduction, references are given to some of the molecular endofullerenes produced by the Kyoto group and others, but some important systems of this kind are omitted, for example the water endofullerene (Murata and co workers) and also the HF and CH4 endofullerenes (Whitby and co workers) 3. The ball and stick graphics in Fig.1 do not depict the chemical structures of these compounds clearly enough. They should be supplemented by ChemDraw-style line structures showing clearly the chemical nature of the orifice and the appended groups. 4. There are a few places where I felt that more references would be appropriate, especially for readers who are not highly conversant with EPR techniques. For example the PEANUT method is not referenced. No reference is given for the lambda value for NO (line after Eq.1). No explicit reference is given for the reported data on related systems (end of first paragraph on page 7). 5. It is not clear until quite late in the discussion that the hyperfine data refers to coupling to the 14N nucleus. 6. A comparison with the observations reported on the similar O2 system (Futagoishi et al.) would have been interesting and enhance the manuscript.

---

## Referee Comment (RC2) · Anonymous Referee #2 · 24 Jun 2020

This paper describes an interesting apporach to the study of paramagnetic NO molecules encapsulated in open-cage C60 fullerenes by pulse EPR methods, from echo-detected, field-swept EPR to nutation and ENDOR experiments. The overall data analysis of g-tensor and hyperfine/quadrupolar couplings is very thorough, presented very clearly and in all depth required. The experimental work is cleverly planned and nicely performed and analyzed. I only have a few comments and points that I would raise: A) The abstract needs some re-writing, compared to the clarity of the paper itself the abstract seems confusing, at least to me. B) At the end of the introduction, the authors state that a smaller g3-value "deduced by an analysis of a CW measurement, necessitated confirmation by pulse ESR experiments, better suited for the study of very broad spectra." I would challenge this view of the difference in CW EPR and

pulse EPR data content. In the case described here, the g3 value of ∼0.2 (instead of ∼0.7 reported now in this paper) wasdeduced by fitting CW EPR spectra (if I understood correctly). In a way, ESE-detected EPR has a built in T2-filter that simplifies the spectra (reducing the broadness of the high-field region). I think the authors should point that out, as this is the realoriginal point they have made here (or, if they think this is wrong, explain, why pulse EPR may be better suited). C) Figure 1 clearly needs to be amended with a chemical structure of the cage molecules - the DFT structures should also be shown from a side view, not only the top view. D) Figure 4 (PEANUT): the authors should explain the three black lines in the graph E) last line on page 6: "level splitting" instead of "spitting" F) Figs. 5 and 6 should be combined in one figure as a) and b) or Fig 6 should be moved into the SI G) Again Figs 7 and 8 should be combined into one figure as a) and b) H) Page 10, beginning of 3.2: Would the authors expect a better resolution at intermediate field values when the frequencies for the ENDOR experiments are varied (through changes in g-strain?)? I) Figures 9, 10, 11: While I am usually all for original presentation of data, in this case I just think that plotting the three spectra in three rows above each other in Fig. 9 (without additional 3-D shift) would make it easier to see the "evolution" of orientation-dependent spectra. Then, Figs. 10 (and 11, I believe) could also be included in this figure.

---

## Author Comment (AC1) · 24 Jul 2020

We are grateful to the referee for the constructive and helpful comments on our manuscript. Below we respond point-by-point to the comments and include suggestions for corresponding revisions of the manuscript.

Referee 1 Malcom Levitt:
*1) In the abstract, it is not very clear that the confining cage is not C60 but an open-cage variant (in fact two variants) of C60.*
Response: The first sentence of the abstract will be rephrased: "... confined in two different modified open C60 derived cages are determined." Further changes will be made in reply to Referee 2, 1).

*2) In the introduction, references are given to some of the molecular endofullerenes produced by the Kyoto group and others, but some important systems of this kind are omitted, for example the water endofullerene (Murata and coworkers) and also the HF and CH4 endofullerenes (Whitby and coworkers).*
Response: Three additional references will be included:
a) K. Kurotobi and Y. Murata, Science, 2011, 333, 613–616 (H2O@C60)
b) A. Krachmalnicoff et al., Nature Chemistry, 2016, 8(10) 953–957 (HF@C60)
c) S. Bloodworth et al., Ang. Chem. Int. Ed., 2019, 58(15) 5038–5043 (CH4O@C60)

*3) The ball and stick graphics in Fig.1 do not depict the chemical structures of these compounds clearly enough. They should be supplemented by ChemDraw-style line structures showing clearly the chemical nature of the orifice and the appended groups.*
Response: Fig. 1 will be changed to display only NO@C60-OH1 with the left part as "side view" of the modified C60 cage represented by true balls-and-sticks together with the caged NO represented by van-der-Waals spheres. The right part will show a "top view" onto the orifice with the orifice atoms represented by van-der-Waals spheres. The caged NO will again also be displayed as van-der-Waals spheres, but the modified C60 cage represented only by sticks. NO@C60-OH3 will be shown as Fig. A1 in the appendix identical to the original Fig. 1b). We attach the new Fig. 1 Left (side view of NO@C60-OH1 as Fig. 1 to this comment.

*4) There are a few places where I felt that more references would be appropriate, especially for readers who are not highly conversant with EPR techniques. For example the PEANUT method is not referenced. No reference is given for the lambda value for NO (line after Eq.1). No explicit reference is given for the reported data on related systems (end of first paragraph on page 7).*
Response: References had been misplaced at several instances and will be given at the appropriate places, e.g. the PEANUT reference (Stoll et al., 1998) was misplaced

at line 47 instead of line 45.

*5) It is not clear until quite late in the discussion that the hyperfine data refers to coupling to the $^{14}N$ nucleus.*
Response: This will be clarified by replacing the sentence starting at line 150 by:
"The frequency position at the low field side of the spectrum and the magnitude of the shift is inconsistent with proton hfi but is indicative for a dominant dipolar $^{14}$N hfi, . . .".

*6) A comparison with the observations reported on the similar O2 system (Futagoishi et al.) would have been interesting and enhance the manuscript.*
Response: This will be incorporated at line 99 after ". . . (200 K) is determined" by:
"The lifting of $^2\pi_x/^2\pi_y$ degeneracy is not unexpected considering the observation of a finite zero-field-splitting for $^3O_2$ in a cage with $C_1$ symmetry (Futagoishi et al., 2017). In this study the potential barrier for librational motions of $^3O_2$ was estimated as 398 cm$^{-1}$, by measuring the shift of its principal ZFS component with respect to the value of the free molecule. The size of this potential barrier is of the same order of magnitude as the one calculated for NO."

[Figure]

**Fig. 1.**

---

## Author Comment (AC2) · 24 Jul 2020

We are grateful to the referee for the constructive and helpful comments on our manuscript. Below we respond point-by-point to the comments and include suggestions for corresponding revisions of the manuscript.

Referee 2

*1) The abstract needs some re-writing, compared to the clarity of the paper itself the abstract seems confusing, at least to me.*
Response: The abstract will be modified for clarification (see also Ref 1, 1)).

*2) At the end of the introduction, the authors state that a smaller g3-value "deduced*

[Figure]

*by an analysis of a CW measurement, necessitated confirmation by pulse ESR experiments, better suited for the study of very broad spectra." I would challenge this view of the difference in CW EPR and pulse EPR data content. In the case described here, the g3 value of 0.2 (instead of 0.7 reported now in this paper) was deduced by fitting CW EPR spectra (if I understood correctly). In a way, ESE-detected EPR has a built in T2-filter that simplifies the spectra (reducing the broadness of the high-field region). I think the authors should point that out, as this is the real original point they have made here (or, if they think this is wrong, explain, why pulse EPR may be better suited).*

Response: The referee is quite right in his remark that because of possible $T_2$ variations in the spectral range the true absorption line shape can be distorted in an FSE spectrum. However, we disagree with the referee about a "simplification", i.e. complete loss of the extreme high-field part, of the spectrum suggested to exist for $g_3$ = 0.225 in the FSE spectra. The $T_2$ determined by a 2-pulse echo sequence does not show any significant shortening towards the high-field end of the observed FSE spectrum, instead $T_2$ increases at temperature used for the FSE spectra with increasing field. Furthermore, the FSE spectrum shows a clear high-field shoulder which is however much too broad for being picked up by typically achievable $B_0$ modulation amplitudes. In addition we would like to point out that $g_3$ = 0.225 corresponds to $B_0$ beyond 3 T, while the cw spectra were recorded only up to 1.5 T. For clarification, a sentence will be added to the introduction at line 26 after ". . . very broad spectra":

"Although a $T_2$ variation as function of the external field can distort the shape of a pulse derived spectrum to some extent, difficulties in detecting extremely broad spectra with virtually absent changes within the typically achievable $B_0$ modulation amplitudes in cw ESR can lead to misinterpretations, in particular if the suggested spectrum extends a factor of two beyond the possible acquisition range."

*3) Figure 1 clearly needs to be amended with a chemical structure of the cage molecules - the DFT structures should also be shown from a side view, not only the*

*top view.*

Response 3): Figure 1 will be modified (see Referee 1, 3)).

*4) Figure 4 (PEANUT): the authors should explain the three black lines in the graph.*

Response: Figure and caption will be modified. The corresponding sentence in the caption will read:

"The dashed vertical lines indicate the expected nutation frequency distributions (Stoll et al., 1998) at the three principal *g* values for NO@C60-OH3 in Table 1 (X band)."

*5) Last line on page 6:"level splitting" instead of "spitting": Thank you for spotting.*

*6) Figs. 5 and 6 should be combined in one figure as a) and b) or Fig 6 should be moved into the SI. Again Figs 7 and 8 should be combined into one figure as a) and b).*

Response: Figures 5 and 6 as well as Figs. 7 and 8 will be combined in one figure, respectively. We also would combine Figs. 2 and 3 as well as Figs. A1 and A2 into one figure, respectively.

*7) Page 10, beginning of 3.2: Would the authors expect a better resolution at intermediate field values when the frequencies for the ENDOR experiments are varied (through changes in g-strain?)?*

Response: Best resolution is expected for van Hove orientations since in our case g matrix axes and hfi axes are expected to be collinear. This is true irrespective of the mw frequency used. For arbitrary orientations it is expected that at 34 GHz a better resolution in ENDOR would be obtained. The corresponding field range, however, was inaccessible for us due to low g values of interest here.

*8) Figures 9, 10, 11: While I am usually all for original presentation of data, in this case I just think that plotting the three spectra in three rows above each other in Fig. 9 (without additional 3-D shift) would make it easier to see the "evolution" of orientation-dependent spectra. Then, Figs. 10 (and 11, I believe) could also be included in this figure.*

Response: We prefer to keep the presentation of Fig. 9, however, will combine it with Fig. 10, but keep Fig. 11 separate.

---

## Author Response (AR1)

Responses to referee comments

We are grateful to both referees for their constructive and helpful comments on our manuscript. Below we respond point-by-point to the comments and list the changes made to the manuscript.

Referee 1 Malcom Levitt:

*1) In the abstract, it is not very clear that the confining cage is not C60 but an open-cage variant (in fact two variants) of C60.*

Response: The first sentence of the abstract has been rephrased:
"… confined in two different modified open $C_{60}$ derived cages are determined."
Further changes have been made in reply to Referee 2, 1).

*2) In the introduction, references are given to some of the molecular endofullerenes produced by the Kyoto group and others, but some important systems of this kind are omitted, for example the water endofullerene (Murata and coworkers) and also the HF and CH4 endofullerenes (Whitby and coworkers).*

Response: Three additional references have been included:
a) K. Kurotobi and Y. Murata, Science, 2011, 333, 613–616 ($H_2O@C_{60}$)
b) A Krachmalnicoff *et al.*, Nature Chemistry, 2016, 8(10) 953–957 ($HF@C_{60}$)
c) S Bloodworth *et al.*, Ang. Chem. Int. Ed., 2019, 58(15) 5038–5043 ($CH_4O@C_{60}$)

*3) The ball and stick graphics in Fig.1 do not depict the chemical structures of these compounds clearly enough. They should be supplemented by ChemDraw-style line structures showing clearly the chemical nature of the orifice and the appended groups.*

Response: Fig. 1 has been changed to display only NO@C60-OH1 with the left part as "side view" of the modified $C_{60}$ cage represented by true balls-and-sticks together with the caged NO represented by van-der-Waals spheres. The right part now shows a "top view" onto the orifice with the orifice atoms represented by van-der-Waals spheres. The caged NO again is also displayed as van-der-Waals spheres, but the modified $C_{60}$ cage represented only by sticks. NO@C60-OH3 is now shown as Fig. A1 in the appendix identical to the original Fig. 1b).

*4) There are a few places where I felt that more references would be appropriate, especially for readers who are not highly conversant with EPR techniques. For example the PEANUT method is not referenced. No reference is given for the lambda value for NO (line after Eq.1). No explicit reference is given for the reported data on related systems (end of first paragraph on page 7).*

Response: References had been misplaced at several instances and are given now at the appropriate places.

*5) It is not clear until quite late in the discussion that the hyperfine data refers to coupling to the 14N nucleus.*

Response: This has been clarified by replacing the sentence starting at line 150 (now line 165) by:
"The frequency position at the low field side of the spectrum and the magnitude of the shift is inconsistent with proton hfi but is indicative for a dominant dipolar $^{14}N$ hfi, …".

*6) A comparison with the observations reported on the similar O2 system (Futagoishi et al.) would have been interesting and enhance the manuscript.*

Response: This has been incorporated at line 99 after "… (200 K) is determined" (now line 108) by:
"The lifting of $^2\pi_x/^2\pi_y$ degeneracy is not unexpected considering the observation of a finite zero-field-splitting for $^3O_2$ in a cage with $C_1$ symmetry (Futagoishi et al., 2017). In this study the potential barrier for librational motions of $^3O_2$ was estimated as 398 $cm^{-1}$, by measuring the shift of its principal ZFS component with respect to the value of the free molecule. The size of this potential barrier is of the same order of magnitude as the one calculated for NO."

Referee 2:

*1) The abstract needs some re-writing, compared to the clarity of the paper itself the abstract seems confusing, at least to me.*

Response: The abstract has been modified for clarification (see also Ref 1, 1)).

*2) At the end of the introduction, the authors state that a smaller g3-value "deduced by an analysis of a CW measurement, necessitated confirmation by pulse ESR experiments, better suited for the study of very broad spectra." I would challenge this view of the difference in CW EPR and pulse EPR data content. In the case described here, the g3 value of 0.2 (instead of 0.7 reported now in this paper) was deduced by fitting CW EPR spectra (if I understood correctly). In a way, ESE-detected EPR has a built in T2-filter that simplifies the spectra (reducing the broadness of the high-field region). I think the authors should point that out, as this is the real original point they have made here (or, if they think this is wrong, explain, why pulse EPR may be better suited).*

Response: The referee is quite right in his remark that because of possible $T_2$ variations in the spectral range the true absorption line shape can be distorted in an FSE spectrum. However, we disagree with the referee about a "simplification", i.e. complete loss of the extreme high-field part, of the spectrum suggested to exist for $g_3 = 0.225$ in the FSE spectra. The $T_2$ determined by a 2-pulse echo sequence does not show any significant shortening towards the high-field end of the observed FSE spectrum, instead $T_2$ increases at temperature used for the FSE spectra with increasing field. Furthermore, the FSE spectrum shows a clear high-field shoulder which is however much too broad for being picked up by typically achievable $B_0$ modulation amplitudes. In addition we would like to point out that $g_3 = 0.225$ corresponds to $B_0$ beyond 3 T, while the cw spectra were recorded only up to 1.5 T. For clarification, a sentence has been added to the introduction at line 26 after "… very broad spectra" (now line 29):

"Although a $T_2$ variation as function of the external field can distort the shape of a pulse derived spectrum to some extent, difficulties in detecting extremely broad spectra with virtually absent changes within the typically achievable $B_0$ modulation amplitudes in cw ESR can lead to misinterpretations, in particular if the suggested spectrum extends a factor of two beyond the possible acquisition range."

In consequence, we have deleted the sentence starting "The FSE technique, …" on line 68 (now lines 76/77).

*3) Figure 1 clearly needs to be amended with a chemical structure of the cage molecules - the DFT structures should also be shown from a side view, not only the top view.*

Response 3): Figure 1 has been modified (see Referee 1, 3)).

*4) Figure 4 (PEANUT): the authors should explain the three black lines in the graph.*

Response: Figure and caption have been modified. The corresponding sentence in the caption no reads:

"The dashed vertical lines indicate the expected nutation frequency distributions (Stoll et al., 1998) at the three principal *g* values for NO@C60-OH3 in Table 1 (X band)."

*5) Last line on page 6:"level splitting" instead of "spitting":* Thank you for spotting (corrected).

*6) Figs. 5 and 6 should be combined in one figure as a) and b) or Fig 6 should be moved into the SI. Again Figs 7 and 8 should be combined into one figure as a) and b).*

Response: Figures 5 and 6 as well as Figs. 7 and 8 have been combined in one figure, respectively. We also have combined Figs. 2 and 3 as well as Figs. A1 and A2 into one figure, respectively. Correspondingly, figures have been renamed: Figs. 2, 3 are now Fig. 2, Fig. 4 is Fig. 3, Figs. 5, 6 are Fig. 4, Figs. 7, 8 are Fig. 5, and Figs. A1, A2 are Fig. A2. However, we would like to point out that Figs. 2, 4, 5, and A2 should be set spanning both columns in the final two column layout of the manuscript.

*7) Page 10, beginning of 3.2: Would the authors expect a better resolution at intermediate field values when the frequencies for the ENDOR experiments are varied (through changes in g-strain?)?*

Response: Best resolution is expected for van Hove orientations since in our case $g$ matrix axes and hfi axes are expected to be collinear. This is true irrespective of the mw frequency used. For arbitrary orientations it is expected that at 34 GHz a better resolution in ENDOR would be obtained. The corresponding field range, however, was inaccessible for us due to low $g$ values of interest here.

*8) Figures 9, 10, 11: While I am usually all for original presentation of data, in this case I just think that plotting the three spectra in three rows above each other in Fig. 9 (without additional 3-D shift) would make it easier to see the "evolution" of orientation-dependent spectra. Then, Figs. 10 (and 11, I believe) could also be included in this figure.*

Response: We prefer to keep the presentation of Fig. 9, however, have combined it with Fig. 10, but keep Fig. 11 separate. Figs. 9, 10 are now Fig. 6, and Fig. 11 is Fig. 7. Again, we would like to point out that Fig. 6 should be set spanning both columns in the final two column layout of the manuscript.

Further changes have been made to correct mistakes and to improve the manuscript.
The $T_2$ values given have been corrected by a factor of 2 (corrections in Figs. 4R, 5L). However, this has no impact on the results since the $T_2$ value have not been interpreted, but only their temperature dependence. The exponential fitting of the temperature dependence remains unaffected. The estimation of libration mode energies (now line 157) has been rephrased and another factor of 2 has been corrected. This again does not change any conclusion since the estimated value is off by an order of magnitude from the experimental observation.
The order of sentences has been changed at some instances to improve readability.
In Fig. 5 left, we have exchanged the 12.5 K data against 10 K data since the $T_2$ fits are significantly more stable due to the better S/N at this temperature.

A manuscript with all changes highlighted in blue is attached.

[revised manuscript text omitted]